# Universal Visuo-Tactile Video Understanding for Embodied Interaction

**Yifan Xie[1], Mingyang Li[1], Shoujie Li[1], Xingting Li[1], Guangyu Chen[2],**
**Fei Ma[3]\*, Fei Yu[3], Wenbo Ding[1]\***

[1] Tsinghua Shenzhen International Graduate School, Tsinghua University
[2] Shenzhen Campus of Sun Yat-sen University
[3] Guangdong Laboratory of Artificial Intelligence and Digital Economy (SZ)

## Abstract

Tactile perception is essential for embodied agents to understand physical attributes of objects that cannot be determined through visual inspection alone. While existing approaches have made progress in visual and language modalities for physical understanding, they fail to effectively incorporate tactile information that provides crucial haptic feedback for real-world interaction. In this paper, we present VTV-LLM, the first multi-modal large language model for universal Visuo-Tactile Video (VTV) understanding that bridges the gap between tactile perception and natural language. To address the challenges of cross-sensor and cross-modal integration, we contribute VTV150K, a comprehensive dataset comprising 150,000 video frames from 100 diverse objects captured across three different tactile sensors (Gel-Sight Mini, DIGIT, and Tac3D), annotated with four fundamental tactile attributes (hardness, protrusion, elasticity, and friction). We develop a novel three-stage training paradigm that includes VTV enhancement for robust visuo-tactile representation, VTV-text alignment for cross-modal correspondence, and text prompt finetuning for natural language generation. Our framework enables sophisticated tactile reasoning capabilities including feature assessment, comparative analysis, scenario-based decision making and so on. Experimental evaluations demonstrate that VTV-LLM achieves superior performance in tactile video understanding tasks, establishing a foundation for more intuitive human-machine interaction in tactile domains.

## 1 Introduction

Touch is a fundamental sensory modality that provides humans with physical information unattainable through vision alone, such as material attributes, surface texture, and compliance. This tactile feedback enables sophisticated physical reasoning and interaction in our environment [1, 2, 3]. While recent advances in vision-language models [4, 5, 6, 7, 8] have demonstrated impressive capabilities in visual reasoning, these models remain fundamentally limited by their inability to perceive tactile attributes, restricting their effectiveness in scenarios requiring physical interaction and reasoning about material characteristics that cannot be reliably inferred from visual cues alone.

Visuo-tactile sensors [9], like GelSight [10], DIGIT [11], and Tac3D [12], have emerged as promising technologies for capturing tactile information, generating image-like representations that encode physical properties such as pressure distribution, surface geometry, and friction characteristics. However, there remains a significant challenge in bridging the domain gap between these tactile representations and natural language understanding. The inherent differences between tactile data

---

*Corresponding Author.

39th Conference on Neural Information Processing Systems (NeurIPS 2025).

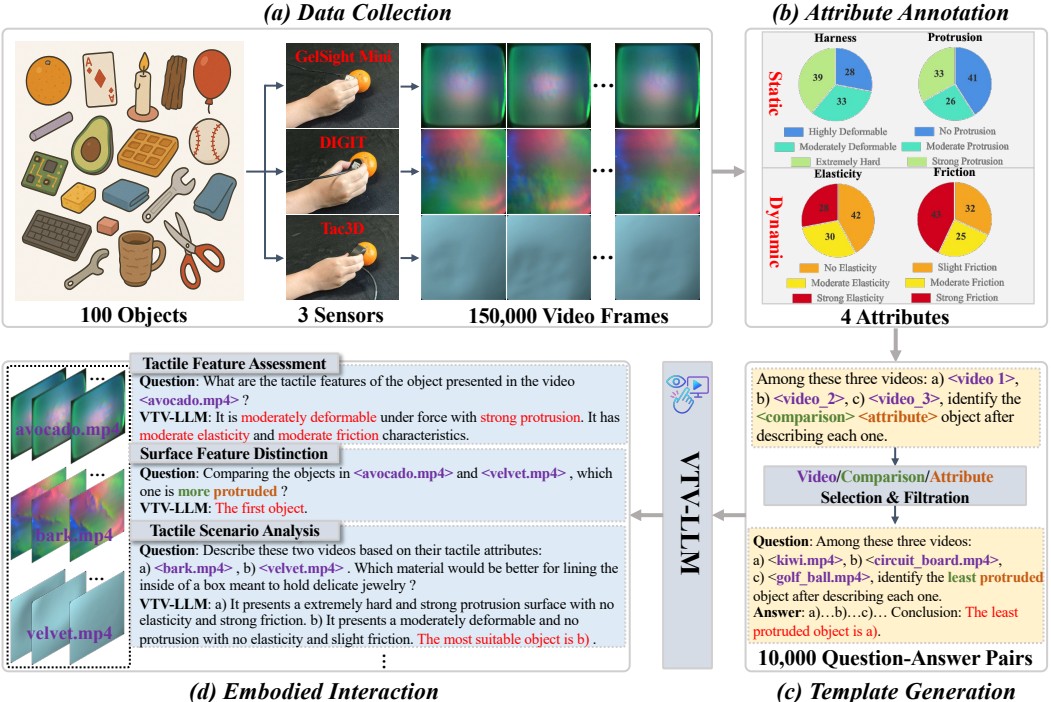

Figure 1: The workflow consists of four key components: (a) Data Collection, which includes 100 diverse objects recorded by 3 different tactile sensors, resulting in 150,000 video frames; (b) Attribute Annotation, where objects are systematically categorized across 4 static and dynamic tactile attributes: hardness, protrusion, elasticity, and friction; (c) Template Generation, which generates 10,000 question-answer pairs using structured templates for various comparative analyses; and (d) Embodied Interaction, demonstrating VTV-LLM's capability to perform tactile feature assessment, surface feature distinction, tactile scenario analysis and so on. Through this integrated approach, VTV-LLM enables multi-modal reasoning about physical attributes that cannot be determined through visual inspection alone, creating a foundation for more sophisticated human-machine interaction in tactile understanding domains.

captured across various sensor types further complicates this integration, as each sensor produces distinct data formats with varying resolutions and physical property encodings.

Existing research on tactile learning has made progress in representation learning [13, 14, 15, 16, 17, 18], but these approaches often focus either exclusively on static attributes or fail to develop comprehensive frameworks that integrate both tactile perception and language understanding. Most critically, they lack the ability to ground tactile perceptions in natural language descriptions and reasoning, which is essential for human-machine communication about physical properties and interactions [19, 20]. Additionally, the temporal dimension of tactile interactions, which captures how surfaces respond to pressing, sliding, and rotational movements, remains underexplored in current approaches, despite containing crucial information about dynamic material attributes.

To address these challenges, we present VTV-LLM, the first multi-modal large language model for universal visuo-tactile video understanding. Our approach treats tactile perception as a cross-modal reasoning problem, where tactile videos are aligned with linguistic descriptions to enable sophisticated reasoning about physical attributes. As illustrated in Fig. 1(d), VTV-LLM supports a diverse range of embodied interaction capabilities, from basic tactile feature assessment to complex comparative analyses and scenario-based decision making. Additionally, we construct the VTV150K dataset, comprising 150,000 video frames collected from 100 common objects across three different tactile sensors. We systematically annotate these videos with four fundamental tactile attributes (hardness, protrusion, elasticity, and friction), creating a structured foundation for tactile reasoning. To bridge the substantial gap between tactile perception and language understanding, we develop a three-stage training paradigm: (1) VTV enhancement through optical flow-guided masking to learn

robust visuo-tactile representations, (2) VTV-text alignment to establish cross-modal correspondence, and (3) text prompt finetuning to optimize natural language generation about tactile attributes.

Our main contributions can be summarized as follows:

- We introduce VTV-LLM, the first multi-modal large language model capable of universal visuo-tactile video understanding, enabling sophisticated embodied reasoning through natural language interaction.

- We contribute VTV150K, a comprehensive dataset of 150,000 visuo-tactile video frames capturing 100 diverse objects across three tactile sensors, annotated with four fundamental tactile attributes.

- We develop a novel three-stage training paradigm that effectively bridges the domain gap between tactile perception and language understanding, providing a valuable reference for future cross-modal learning efforts.

## 2   Related Works

**Tactile Perception**   Tactile perception has evolved significantly from early sensors measuring basic physical properties to sophisticated vision-based systems providing high-resolution contact information. Visuo-tactile sensors [9] such as GelSight [10], DIGIT [11], and Tac3D [12] have garnered widespread attention for their ability to capture detailed contact deformations through elastomeric gels and embedded cameras. These sensors have enabled numerous robotic applications including material classification [21], shape reconstruction [22, 23], and dexterous manipulation tasks [24, 14]. Recent research has focused on developing representation learning approaches for tactile data, progressing from task-specific models [25] to general-purpose representations using self-supervised techniques like contrastive multi-view coding [21] and masked autoencoders [26]. The integration of tactile sensing with other modalities has also emerged as a promising direction, with works like UniTouch [17] dynamically fusing tactile signals with visual and audio data to enhance cross-sensor knowledge transferability, Yu et al. [15] aligned tactile images with vision-language models for object property reasoning, and Fu et al. [16] used a touch-vision-language model for open-vocabulary classification. Unlike prior works, our method processes visuo-tactile video directly and focuses on sophisticated tactile reasoning.

**Self-Supervised Video Representation Learning**   Self-supervised video representation learning has emerged as a critical area for developing robust visual features without manual annotations. VideoMAE [27] pioneered this approach by effectively adapting masked autoencoding strategies to the video domain, demonstrating significant performance improvements across various benchmark tasks. Subsequently, VideoMAEv2 [28] enhanced this framework through the introduction of dual masking mechanisms, which substantially improved computational efficiency while maintaining representational power. Recent advancements in this field have focused on sophisticated optimizations along both temporal and spatial dimensions [29, 30, 31, 32], addressing challenges unique to video understanding such as motion coherence and long-range dependencies. In the tactile domain, Sparsh [18] explored the ability of different existing self-supervised learning methods to characterize in tactile video. Feng et al. [13] utilized the tube masking strategy to process the tactile video. Our method builds upon these foundations by introducing optical flow-guided masking specifically designed for visuo-tactile videos, which addresses the unique challenges of capturing both spatial deformation and temporal dynamics in tactile interactions.

**Multi-Modal Large Language Models**   Multimodal Large Language Models (MLLMs) have transformed AI research by enabling reasoning across textual and visual modalities. Early efforts integrated LLMs as agents for downstream tasks [33, 34, 35]. Later approaches focused on parameter-efficient tuning [36, 37] and instruction tuning [38, 39] to align visual semantics with language. Recent advances have incorporated video processing [40, 41] and diverse sensory inputs [42], enabling applications in robotics [43, 44]. In our work, we present the first visuo-tactile video large language model to bridge the gap between tactile perception and natural language.

# 3 Methods

In this section, we first introduce VTV150K, a large-scale dataset of video-question-answer pairs in Sec. 3.1. Subsequently, we present VTV-LLM, the first visuo-tactile video large language model designed for visuo-tactile video understanding and embodied interaction in Sec. 3.2.

## 3.1 VTV150K

**Overview**  Visuo-tactile sensor technologies suffer from inadequate standardization and significant cross-sensor data discrepancies, which substantially impede the transferability of tactile representation models across different sensing platforms. Existing methods [14, 18, 45, 13] addressing these challenges exhibit notable limitations, as they either neglect the integration of both static and dynamic tactile attributes or fail to incorporate comprehensive visuo-tactile video understanding for embodied interaction.

In this work, we introduce VTV150K, a comprehensive large-scale dataset comprising video-question-answer pairs collected across three diverse visuo-tactile sensors, as illustrated in Fig. 1(a-c). The dataset construction methodology encompasses three sequential stages: data collection, attribute annotation, and template generation. We will delve into the specifics of these stages.

**Data Collection**  To facilitate the grounding of embodied interaction on tactile inputs, we collected a comprehensive dataset comprising 100 common objects, yielding a total of 150,000 visuo-tactile video frames, with each video recorded at 20 FPS and a resolution of 320×320 pixels.

As illustrated in Fig. 1(a), we employed multiple visuo-tactile sensors to ensure style diversity: GelSight mini [10] and DIGIT [11] sensors for capturing high-resolution visuo-tactile information, and Tac3D [12] for measuring deformation force fields. Due to the relatively low resolution of Tac3D, we implemented the cubic spline interpolation algorithm [46] to reconstruct more detailed force field representations.

Data collection was performed manually to address the challenges associated with properly interacting with irregularly shaped objects. For each object, we systematically captured five visuo-tactile videos across different regions using various sensors. Our data collection process consisted of three sequential interactions: (1) normal pressing against the object surface to capture pressure distribution, (2) rotational movement to acquire shear information, and (3) sliding motion to obtain friction characteristics. This multi-interaction approach enables comprehensive tactile information extraction for embodied interaction.

**Attribute Annotation**  To facilitate tactile reasoning, we annotated our dataset across four fundamental static and dynamic tactile attributes as shown in Fig. 1(b). Each attribute was categorized into three distinct levels, with harness classified as highly deformable (28%), moderately deformable (33%), and extremely hard (39%); protrusion categorized as absent (41%), moderate (26%), or strong (33%); elasticity measured as none (42%), moderate (30%), or strong (28%); and friction assessed as slight (32%), moderate (25%), or strong (43%). This structured annotation framework enables comprehensive tactile attribute analysis for downstream reasoning tasks.

**Template Generation**  Template generation facilitates the creation of question-answer pairs for model training. We developed multiple problem templates encompassing various reasoning tasks: tactile feature assessment, surface feature distinction, texture optimal selection and so on. To instantiate these templates, we systematically integrated diverse visuo-tactile video combinations, comparison operators (*e.g.*, "more", "less", "most", "least"), and attribute selectors to generate a comprehensive dataset of 10,000 question-answer pairs. As illustrated in Fig. 1(c), our generation process follows a hierarchical framework: selection, filtration, and structured question formulation with corresponding ground-truth annotations. For more comprehensive details about attribute annotation and template generation, please refer to the Supplementary Material **??**.

## 3.2 VTV-LLM

**Overview**  VTV-LLM aims to serve as a multi-modal framework capable of integrating visual-tactile video data with large language models to facilitate tactile reasoning for embodied interaction.

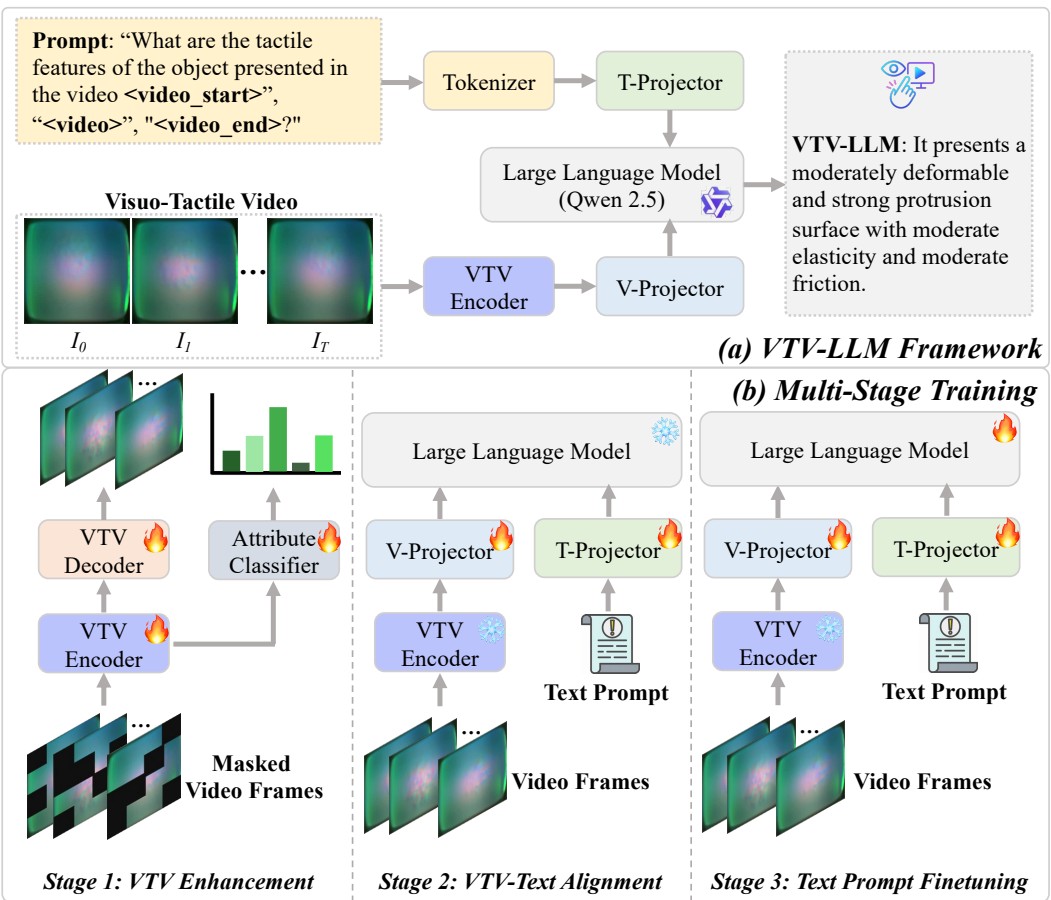

Figure 2: (a) VTV-LLM framework: A multi-modal system integrating visual-tactile video data with large language models to facilitate tactile reasoning for embodied interaction; (b) Multi-Stage Training: It consists of VTV enhancement, alignment between visuo-tactile video and text, and prompt-based finetuning to generate accurate tactile descriptions.

As illustrated in Fig. 2(a), VTV-LLM formulates tactile perception as a cross-modal approach to question answering and descriptive generation. By leveraging the rich sensory information inherent in visuo-tactile video data, VTV-LLM enhances understanding in scenarios traditionally challenging for standard vision-only models, particularly in applications requiring tactile attribute inference.

At the core of VTV-LLM lies a (Qwen 2.5 [4, 5]) that synthesizes complex multi-modal information from visuo-tactile videos, utilizing world knowledge to generate coherent, human-readable descriptions of tactile attributes. In general, a visuo-tactile video can be mathematically represented as a sequence of frames $\mathcal{V} = \{I_t\}_{t=0}^{T}$, where each frame $I_t$ captures both visual and tactile information at timestamp $t$. Initially, high-dimensional features $F_{VTV}$ are extracted from $\mathcal{V}$ using a VTV encoder based on ViT-base architecture [47] adapted from VideoMAE [27, 28]:

$$F_{VTV} = f_{\text{enc}}(V) = \text{ViT}\left(\{\text{Patch}(I_t) + \text{TE}(t)\}_{t=0}^{T}\right), \qquad (1)$$

where $\text{Patch}(\cdot)$ denotes the patch embedding operation and $\text{TE}(t)$ represents temporal embeddings. These features are then processed through a visual projector $f_{V-proj}$ consisting of two linear layers with a GELU activation function [48] in between to produce the visual embedding $E_V$:

$$E_V = f_{V-proj}(F_{VTV}) = W_2 \cdot \text{GELU}(W_1 \cdot F_{VTV} + b_1) + b_2, \qquad (2)$$

where $W_1, W_2$ are learnable weight matrices and $b_1, b_2$ are bias terms. Concurrently, the textual prompt is tokenized and processed through LLM's text projector to produce text embedding $E_T$. For effective multi-modal reasoning, we introduce special tokens <video_start>, <video>, and

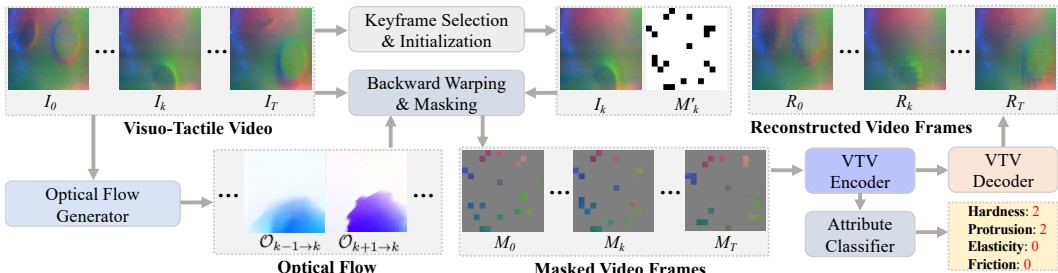

Figure 3: Training pipeline of VTV enhancement.

<video_end> to denote the beginning, content and end of the visuo-tactile video in the input sequence. These tokens serve as anchors for the model to properly align visual information with textual understanding during the inference process.

Given these aligned representations, the large language model $f_{LLM}$ performs reasoning to generate a response $A$ describing tactile attributes:

$$A = f_{LLM}(E_V, E_T) = \text{Qwen}(\text{Concat}([E_V; E_T])). \tag{3}$$

Given the complexity of integrating visuo-tactile information with language representations, we implement a staged training approach to develop our framework. As shown in Fig. 2(b), VTV-LLM adopts a three-stage training paradigm encompassing VTV enhancement, VTV-text alignment, and text prompt finetuning. This structured progression enables the model to first learn robust tactile-visual representations, then align them with textual descriptions, and finally optimize response generation, enhancing VTV-LLM's capability for cross-modal understanding and tactile reasoning. In the following, we describe each of these stages in detail.

**VTV Enhancement**  Existing multi-modal LLMs predominantly process natural images via un-modified Vision Transformer (ViT) encoders [47]. However, our research addresses visuo-tactile inputs, which exhibit fundamentally different characteristics from natural images, thus necessitating specialized fine-tuning to extract meaningful representations.

Furthermore, the temporal nature of our video data introduces challenges not present in static images. Unlike images, videos possess an inherent time dimension characterized by temporal redundancy and inter-frame correlations, requiring robust video representation methodologies. While VideoMAE [27, 28] offers a powerful masked video autoencoder with an asymmetric encoder-decoder architecture utilizing tube masking, this approach assumes minimal motion across large frame regions. This assumption proves problematic for visuo-tactile videos, which typically exhibit significant motion patterns. Direct application of tube masking to such inputs risks substantial information leakage, wherein the model can trivially reconstruct masked segments using visible tokens from temporally adjacent frames, which is a critical concern in masked video pre-training. To address these limitations, we propose a novel training pipeline specifically designed for visuo-tactile video representation, as illustrated in Fig. 3.

Given the visuo-tactile video sequence $\mathcal{V} = \{I_t \in \mathbb{R}^{H \times W \times C}\}_{t=0}^T$, where each frame $I_t$ encodes both visual and tactile information at timestamp $t$ with spatial dimensions $H \times W$ and $C$ channels, we propose selecting the middle frame as the keyframe. This selection is motivated by empirical observations that the middle frame typically exhibits the maximum contact surface area, facilitating more robust optical flow warping in subsequent processing stages. For keyframe mask initialization, conventional binarization approaches [49] significantly degrade the spatial continuity of object surfaces, compromising the fidelity of the reconstructed tactile information. Therefore, we introduce a Gaussian mixture model [50] to obtain the keyframe mask. For the keyframe $I_k$, we formulate a probabilistic mask using localized Gaussian functions. We select a set of $N = \lceil \alpha \cdot HW/\beta^2 \rceil$ sampling points $\{p_i\}_{i=1}^N$ distributed across the frame, where $\alpha \in (0, 1)$ controls density and $\beta$ is the sampling grid size. Each point $p_i$ generates a Gaussian kernel $G_i(x, y) = \exp\left(-\frac{(x-p_{i_x})^2 + (y-p_{i_y})^2}{2\lambda^2}\right)$ with scale parameter $\lambda$. The final keyframe mask is defined as $M_k' = \min\left(1, \sum_{i=1}^N G_i\right)$, creating

a continuous-valued mask that preserves spatial structure while enabling controlled sparsity for subsequent processing.

Additionally, we employ dense motion estimation across the visuo-tactile video $\mathcal{V}$ using the RAFT architecture [51]. We compute bidirectional optical flow fields between consecutive frames to capture the continuous deformation patterns throughout the interaction process. For each adjacent frame pair, we define the forward flow field $\mathcal{O}_{t \to t+1} = \text{RAFT}(I_t, I_{t+1})$. Each flow field $\mathcal{O}_{t \to t+1} \in \mathbb{R}^{H \times W \times 2}$ encodes pixel-wise displacement vectors $(u_{x,y}, v_{x,y})$ for every spatial location $(x, y)$, mapping positions from frame $I_t$ to their corresponding locations in frame $I_{t+1}$. The complete set of optical flows $\Phi$ for the sequence is formulated as:

$$\Phi = \bigcup_{t=0}^{k-1} \{\mathcal{O}_{t \to t+1}\} \cup \bigcup_{t=k+1}^{T} \{\mathcal{O}_{t \to t-1}\}. \tag{4}$$

This bidirectional flow representation tracks visuo-tactile features throughout the interaction, supporting warping operations and masked frame generation. We apply spatial normalization before flow computation to ensure scale invariance across different sequences.

After that, we utilize the backward warping [52, 53] to generate the temporal consistent masking map based on the keyframe and mask the corresponding video frames. The masked visuo-tactile frames $\mathcal{V}_m = \{M_t\}_{t=0}^{T}$ are fed into the VTV encoder-decoder architecture for reconstruction using the mean squared error loss [27, 28]. We also incorporate an attribute classifier to predict tactile attributes (hardness, protrusion, elasticity, and friction) using the cross-entropy loss [54]. Our total loss function combines both the reconstruction loss and the attribute classification loss, enabling simultaneous optimization of visuo-tactile reconstruction quality and tactile attribute classification accuracy.

**VTV-Text Alignment**  In the VTV-text alignment stage, we focus on establishing cross-modal alignment between video and language representations. With the pretrained VTV Encoder from stage 1, we introduce both V-Projector and T-Projector modules while keeping the Large Language Model frozen. This stage leverages our initial constructed VTV150K dataset to bridge the representational gap between visual and textual modalities. The V-Projector maps video embeddings from the VTV Encoder into the language model's embedding space, while the T-Projector processes corresponding text prompt representations. By training these projection modules exclusively while freezing other components, we establish foundational cross-modal understanding, enabling the model to associate visual content with appropriate textual descriptions. This alignment is critical for downstream video understanding and description tasks as it creates a shared semantic space between the video frames and natural language.

**Text Prompt Finetuning**  In the text prompt finetuning stage, we enhance the model's capacity to respond accurately to textual prompts about video content by implementing supervised fine-tuning across multiple components. The V-Projector, and T-Projector are jointly fine-tuned along with the LLM. Unlike previous stages where the LLM remained frozen, this stage employs parameter-efficient techniques [37, 36] to fine-tune the language model using 10,000 newly generated question-answer pairs. These pairs are created using the same template generation approach as our VTV150K dataset, featuring diverse video understanding tasks. By generating new data rather than reusing subsets, we significantly increase training diversity and model robustness. This end-to-end optimization enables the model to generate more coherent, accurate, and contextually relevant responses to text prompts about video content. The supervised nature of this phase significantly improves the model's ability to comprehend complex video scenes and produce natural language descriptions that align with human expectations. This final stage integrates the previously aligned representations into a cohesive multi-modal understanding system, culminating in enhanced video-language capabilities.

## 4 Experiments

### 4.1 Setup

Our experiments utilize the proposed VTV150K dataset for both training and evaluation protocols. The training process follows our three-stage paradigm: Stage 1 employs multi-sensor visuo-tactile videos with their corresponding attribute annotations for representation learning. For Stage 2 and 3,

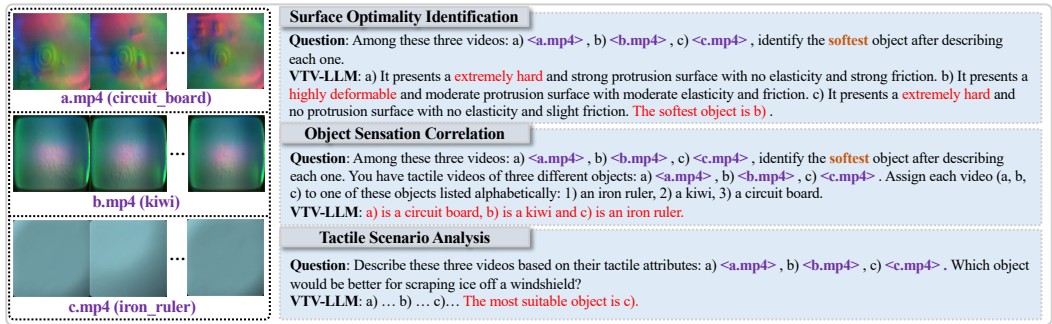

Figure 4: Several task examples from the proposed VTV150K along with predictions from VTV-LLM.

Table 1: Performance comparison of VTV-LLM-7B against seven state-of-the-art methods on the VTV150K dataset. The evaluation covers different tasks, with results reported in percentages (%) and the boldface indicates the best performance.

| Models | Hardness | Protrusion | Elasticity | Friction | Combined | SFD | SOI | OSC | TSA | Average |
|---|---|---|---|---|---|---|---|---|---|---|
| GPT-4o [55] | 34.7 | 32.6 | 32.6 | 18.7 | 2.1 | 40.9 | 38.4 | 16.6 | 36.0 | 28.0 |
| Gemini-2.5-Pro-Exp [56] | 36.2 | 34.7 | 39.1 | 21.0 | 4.3 | 42.6 | 29.4 | 18.5 | 40.0 | 29.5 |
| LLaVA-OneVision-7B [57] | 27.5 | 32.6 | 26.0 | 20.2 | 0.7 | 40.9 | 28.2 | 11.7 | 30.0 | 24.2 |
| LLaVA-Video-Qwen2-7B [58] | 30.4 | 29.7 | 28.9 | 18.1 | 2.1 | 33.6 | 29.4 | 17.2 | 36.0 | 25.0 |
| InternVL2.5-VL-8B [59] | 18.1 | 23.9 | 21.0 | 13.7 | 0.0 | 24.5 | 17.9 | 11.1 | 24.0 | 17.1 |
| VideoLLaMA3-7B [41] | 15.2 | 21.7 | 14.4 | 10.8 | 0.0 | 11.4 | 12.8 | 7.4 | 20.0 | 12.6 |
| Qwen2.5-VL-7B [60] | 25.3 | 28.9 | 17.3 | 15.9 | 1.4 | 22.9 | 28.2 | 16.0 | 30.0 | 20.6 |
| VTV-LLM-7B (Ours) | **73.9** | **75.0** | **67.3** | **56.5** | **35.6** | **71.3** | **57.6** | **43.2** | **64.0** | **60.4** |

we utilize two independently generated sets of 10,000 question-answer pairs to prevent data leakage between stages. To evaluate model performance, we create a separate test set comprising 600 question-answer pairs for novel objects not present in the training data, ensuring comprehensive coverage across various tactile reasoning tasks. Our LLM backbone is based on Qwen 2.5 [4, 5], experimenting with three model variants (3B, 7B, and 14B parameters). All experiments are conducted on 4 NVIDIA RTX 6000 Ada GPUs. Additional implementation details and hyperparameter configurations are provided in the Supplementary Material **??**.

## 4.2 Results

To verify the effectiveness of our VTV-LLM, we compare it against two strong proprietary models, such as GPT-4o [55] and Gemini-2.5-Pro-Exp [56], as well as five open-source video-based VLMs, including LLaVA-OneVision-7B [57], LLaVA-Video-Qwen2-7B [58], InternVL2.5-VL-8B [59], VideoLLaMA3-7B [41] and Qwen2.5-VL-7B [60]. Since most of the video-based VLM models have parameters around 7B, we only use the VTV-LLM-7B model for fair comparison. To guarantee the robustness of the experimental results, we report the average results of the triplicate test with random seeds.

Our first experiment focuses on tactile feature assessment, which evaluates the model's ability to perceive and describe physical sensory attributes of objects in visuo-tactile videos. As illustrated in Fig. 1(d), when presented with a visuo-tactile video and a question prompt, VTV-LLM generates descriptions of the four key tactile attributes. The results presented in Tab. 1 demonstrate that our method consistently outperforms all baseline models across both individual attribute and combined attribute settings. The performance gap is particularly notable in the combined attribute setting, which we attribute to our three-stage training paradigm that effectively bridges the domain gap between tactile perception and natural language understanding.

In addition, we conduct high-level tactile reasoning experiments, including surface feature distinction (SFD), surface optimality identification (SOI), object sensation correlation (OSC), and tactile scenario analysis (TSA). SFD involves comparing tactile qualities between objects to determine relative differences, SOI entails analyzing multiple surfaces to determine which exhibits the highest degree of a particular quality, OSC aims at relating tactile perceptual information to the identity of a

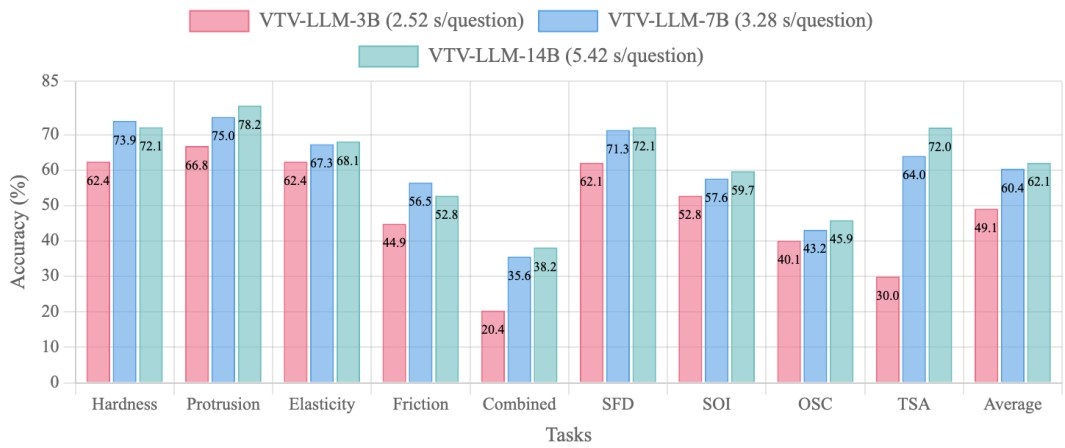

Figure 5: Performance comparison of VTV-LLM on the different parameters.

Table 2: Ablation study on VTV encoder settings using the VTV-LLM-7B model.

| Settings | SFD | SOI | OSC | TSA | Average |
|---|---|---|---|---|---|
| VideoMAE (w/o train) | 37.5 | 29.7 | 8.5 | 16.0 | 22.9 |
| VideoMAE (w/ train) | 52.4 | 46.1 | 28.3 | 38.0 | 41.2 |
| Ours (w/o cls) | 62.2 | 48.7 | 40.1 | 55.0 | 51.5 |
| Ours | **71.3** | **57.6** | **43.2** | **64.0** | **59.0** |

Table 3: Ablation study on three-stage training paradigm settings using the VTV-LLM-7B model.

| Settings | SFD | SOI | OSC | TSA | Average |
|---|---|---|---|---|---|
| w/o stage 2 | 58.1 | 50.0 | 35.2 | 60.0 | 50.8 |
| w/o stage 3 | 50.8 | 42.3 | 29.0 | 52.0 | 43.5 |
| Same dataset | 61.4 | 53.8 | 33.9 | 58.0 | 51.7 |
| Ours | **71.3** | **57.6** | **43.2** | **64.0** | **59.0** |

particular real-world object, and TSA addresses applying haptic knowledge to real-world situations that require physical reasoning. It is worth noting that the TSA task is not included in the training set. The qualitative results presented in Fig. 1(d) and Fig. 4 demonstrate that VTV-LLM can generate reasonable outputs. The quantitative experimental results in Tab. 1 further confirm that VTV-LLM achieves superior performance across these complex reasoning tasks, highlighting its potential for embodied interaction.

## 4.3 Ablation Studies

**LLM Backbone**    To examine the effect of model scale on visuo-tactile understanding, we compare different parameter sizes of our LLM backbone. Fig. 5 shows performance results for VTV-LLM using three Qwen 2.5 variants (3B, 7B, and 14B parameters). We observe consistent performance improvements with increasing model size. This improvement is most significant for complex reasoning tasks like TSA, indicating larger models better integrate cross-modal information. However, larger models also require substantially more computation time during inference.

**VTV Encoder**    We conduct an ablation study on our VTV encoder design as shown in Tab. 2. Baseline VideoMAE [27, 28] without training achieves only 22.9% average performance, while training with our VTV150K dataset improves it to 41.2%. Our method without the attribute classifier reaches 51.5%, showing the effectiveness of our optical flow-guided masking strategy. The full method with the attribute classifier further improves to 59.0%, confirming that joint reconstruction and attribute classification significantly enhances tactile understanding.

**Three-Stage Training Paradigm**    Tab. 3 validates our three-stage training paradigm through ablation studies. Removing stage 2 (VTV-text alignment) drops average performance to 50.8%, while omitting stage 3 (text prompt finetuning) causes a steeper decline to 43.5%. Using identical datasets across stages also underperforms at 51.7%, confirming that independent datasets for each stage significantly improve model robustness.

## 5   Conclusion

In this work, we presented VTV-LLM, the first multi-modal large language model for universal visuo-tactile video understanding. We contributed VTV150K, a comprehensive dataset of visuo-tactile videos across multiple sensors, and developed a novel three-stage training paradigm that effectively bridges the gap between tactile perception and natural language. Experimental results demonstrate that VTV-LLM consistently outperforms state-of-the-art methods across various tactile reasoning tasks, establishing a foundation for more intuitive human-machine interaction in embodied domains.

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
