# OpenReview forum: "Universal Visuo-Tactile Video Understanding for Embodied Interaction"
_NeurIPS.cc/2025/Conference — NeurIPS 2025 poster_

### Official Review · Reviewer_9p99 · 2025-07-01

**Clarity:** 4
**Significance:** 3
**Originality:** 3
**Rating:** 5
**Confidence:** 4

**Summary:**

The paper introduces a large-scale dataset of visuo-tactile video frames captured from 100 objects across three tactile sensors with tactile attributes annotations. It also presents a multi-modal large language model, VTV-LLM, for understanding visuo-tactile videos. The model was trained in three carefully designed stages, to effectively bridge the gap between tactile perception and natural language understanding. The proposed framework enables physical attributes reasoning that cannot be achieved through visual inspection alone. Experimental results show that VTV-LLM achieves superior performance in tactile reasoning tasks over seven state-of-the-art methods on the newly introduced dataset.

**Questions:**

- During the annotation process, what is level of variability of classification levels among three independent annotators? In other words, is it highly subjective and dependent on annotator bias?
- In the question-answer pairs, the success rate of a random guess varies for different tasks, e.g., 50% for SFD and 33% for SOI. Is there any explanation on why some of the open-source models perform much worse than random guesses in Tab. 1?

**Ethical Concerns:**

["NO or VERY MINOR ethics concerns only"]

**Final Justification:**

This paper contributes a comprehensive large-scale dataset of video-question-answer pairs across diverse visuo-tactile sensors. During the rebuttal stage, the concerns on classification levels of tactile attributes and subjectivity of manual annotations were clarified. Reasonable explanations were provided for the significantly below-random performance of some open-source models. Therefore, I recommend acceptance of the paper.

**Limitations:**

Yes.

**Paper Formatting Concerns:**

Nil

**Quality:**

3

**Strengths And Weaknesses:**

Strengths:
- It contributes a comprehensive large-scale dataset of video-question-answer pairs across diverse visuo-tactile sensors.
- The paper provides clear descriptions of dataset construction and model training framework.
- Extensive experiments were conducted for performance comparison and ablation studies.

Weaknesses:
- The classification levels for tactile attributes are coarse-grained, lacking fine-grained precision.
- The manual annotations might be subjective.

---

> ### Author Rebuttal · Authors · 2025-07-25
>
> We thank reviewer for the constructive comments. We provide our feedbacks as follows.
>
> **Q1: The classification levels for tactile attributes are coarse-grained, lacking fine-grained precision.**
>
> **A1**: Thanks for this comment. We understand the reviewer's expectation for finer-grained classification, but our three-level classification design is a reasonable choice based on sufficient theoretical considerations and practical validation. Research in psychology and neuroscience shows that human perception of tactile attributes naturally exhibits discrete and hierarchical characteristics [1][2]. **Our three-level classification (0-1-2) design aligns with the natural boundaries of human tactile cognition, rather than artificially imposing overly fine divisions.**
>
>
> Supplementary Material A.3.1 details our annotation process: *"The annotation process involved three independent annotators evaluating tactile videos of the objects, with final classifications determined by majority consensus."* **Three-level classification ensures high inter-annotator consistency, while overly fine classifications would increase subjectivity and reduce annotation quality.**
>
> Table 1's experimental results show that even under three-level classification, the existing strongest baseline models perform significantly lower than our method across all attributes. **This indicates that the current challenge lies in establishing effective cross-modal understanding, not in insufficient classification granularity.** The Tactile Scenario Analysis (TSA) task shown in Figure 8 indicates that actual embodied interaction focuses more on qualitative decision support rather than precise numerical measurements. For example, questions like "Which object is better for scraping ice?" require categorical judgments rather than continuous values.
>
> Moreover, our three-stage training paradigm (Figure 2 (b)) has good scalability. If finer classification is needed in the future, we can build upon the current framework by adding classification levels or introducing continuous value regression tasks.
> ***
> **Q2: The manual annotations might be subjective. During the annotation process, what is level of variability of classification levels among three independent annotators? In other words, is it highly subjective and dependent on annotator bias?**
>
> **A2**: We provide clear definitions based on objective physical phenomena for each tactile attribute. For elasticity, level 0 represents no obvious rebound, level 1 denotes slow partial recovery, and level 2 denotes rapid full recovery. **These definitions are based on observable physical phenomena, significantly reducing the space for subjective judgment.**
> Moreover, Supplementary Material A.3.1 reports a detailed inter-annotator agreement analysis. We use Fleiss' Kappa coefficient to quantify annotation consistency (Overall: 0.78). According to the standard, 0.61-0.80 indicates "substantial agreement," and 0.81-1.00 indicates "almost perfect agreement," showing that our annotation quality is excellent.
> During the annotation process, we conducted quality checks every 10 completed objects. **Throughout the annotation process, Kappa coefficients remained between 0.75 and 0.82, showing good stability.** We will further clarify related information in the final version.
>
> ***
> **Q3: In the question-answer pairs, the success rate of a random guess varies for different tasks, e.g., 50% for SFD and 33% for SOI. Is there any explanation on why some of the open-source models perform much worse than random guesses in Tab. 1?**
>
> **A3**: Thank you for this insightful question. The significantly below-random performance of some open-source models in Table 1 can be explained by several key factors:
>
> Firstly, the open-source video-language models (e.g., InternVL2.5-VL-8B achieving only 17.1% average) were primarily trained on natural video datasets containing everyday scenes, objects, and activities. As stated in Section 3.1, **visuo-tactile videos exhibit fundamentally different characteristics from natural images, with unique properties like pressure distributions, surface deformations, and contact dynamics that are absent in conventional video datasets.**
>
> Secondly, **rather than making random errors, these models appear to systematically misinterpret tactile sensor outputs.** For instance, they might consistently confuse high-contrast regions in tactile images (which often indicate surface deformations) with visual features like shadows or textures. This leads to consistently wrong answers rather than random guesses, explaining the below-random performance.
>
> Moreover, as mentioned in Section 3.1, our dataset includes three different sensors (GelSight Mini, DIGIT, Tac3D) with distinct data formats, varying resolutions, and physical property encodings. **Models without tactile-specific training struggle with this multi-sensor variability, leading to consistent misclassification patterns.**
>
> This systematic failure actually validates our core contribution - **the necessity of specialized tactile training paradigms rather than simply applying existing video-language models to tactile domains.**
> We will add more discussion in the final version.
> ***
> [1] Liu, Huaping, et al. "Embodied tactile perception and learning." Brain Science Advances 6.2 (2020): 132-158.
>
> [2] Packheiser, Julian, et al. "A systematic review and multivariate meta-analysis of the physical and mental health benefits of touch interventions." Nature human behaviour 8.6 (2024): 1088-1107.

---

> ### Comment · Area_Chair_HA1p · 2025-08-03
>
> Dear Reviewer,
>
> Could you please check if the authors’ rebuttal adequately addresses your concerns? If so, kindly acknowledge the rebuttal and provide any additional comments. If not, it would be greatly appreciated it if you could engage in a discussion with the authors. Your input at this stage is essential to the review process. Thank you very much for your time and effort!
>
> AC

---

> ### Author Response · Authors · 2025-08-05
> **To Reviewer 9p99**
>
> Dear Reviewer 9p99,
>
> Thanks again for your diligent effort in reviewing our submission. We have carefully addressed the concerns raised. As the discussion phase deadline is approaching, we sincerely hope you can consider positively recommending our work if your concerns are solved. If you still have further comments/suggestions, please don't hesitate to let us know.
>
> Best regards, Authors

---

> > ### Comment · Reviewer_9p99 · 2025-08-05
> >
> > Thank you for the thorough answers and clarifications. Since all my questions have been well addressed, I will raise my overall score.

---

> > > ### Author Response · Authors · 2025-08-05
> > >
> > > Thank you very much for your feedback. We are glad that our rebuttal effectively addressed your concerns. We will incorporate the revised content into the final version.

---

> ### Comment · Area_Chair_HA1p · 2025-08-06
>
> Dear Reviewer,
>
> Please remember to submit the *Mandatory Acknowledgement*. Thank you!
>
> AC

---

### Official Review · Reviewer_Q6gv · 2025-07-02

**Clarity:** 2
**Significance:** 3
**Originality:** 2
**Rating:** 4
**Confidence:** 4

**Summary:**

This paper proposes VTV-LLM, the first multi-modal large language model that supports universal visuo-tactile video understanding. The author constructed VTV150K dataset, covering 150k frames of videos from 100 common objects collected by three sensors, annotated with four tactile attributes: hardness, convexity, elasticity, and friction. To achieve efficient multi-modal fusion, the paper designed a three-stage training paradigm: visual tactile representation enhancement, video and text alignment, and text prompt fine-tuning.

**Questions:**

Is it suitable for real-world dexterous manipulation tasks that demand real-time performance, or is it limited to tactile reasoning applications?

**Ethical Concerns:**

["NO or VERY MINOR ethics concerns only"]

**Final Justification:**

This paper proposes VTV-LLM, the first multi-modal large language model that supports universal visuo-tactile video understanding. The author constructed VTV150K dataset, covering 150k frames of videos from 100 common objects collected by three sensors, annotated with four tactile attributes: hardness, convexity, elasticity, and friction. To achieve efficient multi-modal fusion, the paper designed a three-stage training paradigm: visual tactile representation enhancement, video and text alignment, and text prompt fine-tuning. During the discussion, most of raised concerns have been addressed. The reviewer believes this work presents incremental improvements on both the model and the dataset, so finally recommending BA.

**Limitations:**

Yes

**Paper Formatting Concerns:**

No Paper Formatting Concerns.

**Quality:**

2

**Strengths And Weaknesses:**

**Strength:**

1. This work represents the first systematic attempt to develop a dedicated masking strategy specifically for tactile video data.
2. The dataset proposed in this work represents good contribution to the field.


**Weeknesses:**

1. As one of the similar works to Octopi [1]—the first tactile video understanding model—this paper should provide a detailed comparative analysis highlighting both its differences from and advantages over Octopi. Furthermore, compared to models like TVL [2] that only accept single-frame tactile image inputs, how much benefit does video-based tactile understanding actually provide? A rigorous quantitative comparison would better demonstrate the value of temporal modeling in tactile perception.
2. This work proposes an optical flow-guided video reconstruction strategy for tactile videos, which specifically masks deformation regions. However, unlike natural videos, tactile videos exhibit highly localized and fine-grained deformations. Masking 90% of the area may leave insufficient effective information in the remaining regions, potentially causing the mask modeling to overfit. The authors should provide examples of masked images and reconstructed results, and conduct an ablation study on masking ratios to validate the approach. Directly transferring parameters from natural video models may not be optimal for tactile video reconstruction.
3. A key potential of combining tactile sensing with large language models is the ability to achieve zero-shot tactile perception and understanding. To highlight the generalization capability and practical value of their approach, the authors should include experimental results that demonstrate the model’s performance in zero-shot scenarios.
4. I commend the authors for their efforts in collecting the dataset and conducting detailed annotations. However, the dataset’s limited scale—covering only 100 objects with relatively simple interaction patterns—may restrict its applicability and generalization potential in real-world embodied interaction settings.
5. The proposed dataset incorporates Tac3D sensor data with a 20×20 resolution force field alongside high-resolution tactile images.  However, the paper does not clearly explain how this data is combined with high-resolution tactile images. Figures 1 and 7 display two different input forms of Tac3D data (interpolated images and force fields), but it remains unclear which one is actually used—or whether these represent different task settings. What is the size after interpolation? Additionally, is the Tac3D data processed into image-form inputs rather than the original point-cloud-like format? If so, might this approach fail to fully leverage the fine-grained capabilities of Tac3D data in perceiving force?
6. The claim that the method "maintains strong performance across these heterogeneous input modalities" is not sufficiently convincing when relying solely on Figure 7. More comprehensive experimental results should be provided to substantiate this conclusion.


[1] Yu S, Lin K, Xiao A, et al. Octopi: Object property reasoning with large tactile-language models[J]. arXiv preprint arXiv:2405.02794, 2024.
[2] Fu, Letian, et al. "A Touch, Vision, and Language Dataset for Multimodal Alignment." *International Conference on Machine Learning*. PMLR, 2024.

---

> ### Author Rebuttal · Authors · 2025-07-27
>
> We thank reviewer for the constructive comments. We provide our feedbacks as follows.
>
> **Q1: ...this paper should provide a detailed comparative analysis highlighting both its differences from and advantages over Octopi...**
>
> **A1**: Compared to Octopi, our VTV-LLM offers significant improvements in three key areas. First, our VTV150K dataset is **four times larger**, containing 150,000 video frames across 100 objects versus Octopi's PHYSICLEAR dataset. Second, we use **three different tactile sensors** rather than just one GelSight sensor, enabling better generalization across sensor types. Third, we annotate **more tactile properties** for comprehensive physical reasoning tasks.
>
> Technically, **we propose a three-stage training paradigm with optical flow-guided masking for tactile videos, while Octopi uses a basic approach without temporal processing.** Our ablation studies (Table 2) show clear benefits: the full VTV encoder achieves 59.0% average performance versus VideoMAE's 41.2%. Our three-stage interaction design (pressing for pressure, rotation for shear, sliding for friction) captures essential temporal information that single frames cannot provide. This produces significant gains: VTV-LLM achieves 60.4% average performance, substantially outperforming the best baseline's 29.5%.
> To validate temporal modeling's importance in tactile perception, we used Octopi's data reading method (selecting 5 frames) and trained with the same dataset and pipeline. **Results in the table below confirm the importance of tactile video understanding, especially for dynamic attributes.**
> |                    | Hardness | Protrusion | Elasticity | Friction | Combined | SFD  | SOI  | OSC  | TSA  | Average |
> |--------------------|----------|------------|------------|----------|----------|------|------|------|------|---------|
> | Selecting 5 frames | 62.6     | 60.7       | 41.3       | 31.7     | 10.8     | 58.1 | 47.5 | 27.1 | 51.0 | 43.4    |
> | Ours               | **73.9**     | **75.0**       | **67.3**       | **56.5**     | **35.6**     | **71.3** | **57.6** | **43.2** | **64.0** | **60.4**    |
>
> Moreover, regarding direct comparison with TVL, we need to clarify why fair comparison is not feasible. TVL processes single-frame static tactile images, while our approach handles continuous video sequences of dynamic interactions. **The datasets differ significantly**: TVL contains 43,741 static image-touch pairs with 254 tactile adjectives for classification, whereas our VTV150K features video sequences with a structured four-attribute three-level annotation system. **The evaluation tasks also differ**: TVL focuses on classification accuracy while we assess complex multi-modal reasoning like scenario analysis. These fundamental differences make direct comparison both unfair and meaningless, as they address different sub-problems in tactile understanding. We will include this discussion in the final version.
>
> ***
> **Q2: ...Masking 90% of the area may leave insufficient effective information in the remaining regions…**
>
> **A2**: While high masking ratios may seem problematic, our approach has strong theoretical and experimental support.
> VideoMAE [1] research shows that videos contain significant temporal redundancy, with consecutive frames changing slowly and being highly correlated. This allows excellent reconstruction even at 0.9-0.95 masking ratios, as models can infer masked content from limited visible tokens.
> **Tactile videos have even stronger structural advantages than natural videos. Deformations are localized yet constitute important tactile features, and tactile interactions follow predictable physical laws.** Our optical flow-guided masking strategy (Figure 3) preserves key deformation information consistently, while multi-frame sequences provide rich spatiotemporal context even when single frames are limited.
>
> We conducted dedicated masking ratio ablation experiments specifically for tactile videos.
> Results can be found in the following table, where lower performance at 0.7 masking ratio, optimal performance at 0.9. This proves that **high masking ratio provides sufficient challenge for effective representation learning without causing information insufficiency.**
> Figure 3 demonstrates accurate reconstruction of key tactile features, with additional examples planned for the final version.
> |                    | Hardness | Protrusion | Elasticity | Friction | Combined | SFD  | SOI  | OSC  | TSA  | Average |
> |--------------------|----------|------------|------------|----------|----------|------|------|------|------|---------|
> | masking ratio=0.7  | 68.4     | 61.9       | 58.2       | 45.3     | 14.8     | 59.7 | 43.7 | 38.6 | 49.0 | 48.8    |
> | masking ratio=0.8  | 73.7     | **77.3**       | 66.1       | 54.3     | 28.8     | 70.5 | 52.2 | 40.3 | 59.0 | 58.0    |
> | masking ratio=0.9  | **73.9**     | 75.0       | **67.3**       | **56.5**     | **35.6**     | 71.3 | **57.6** | **43.2** | **64.0** | **60.4**    |
> | masking ratio=0.95 | 72.4     | 76.8       | 62.4       | 48.9     | 30.4     | **72.1** | 51.4 | 40.1 | 60.0 | 57.1    |
>
> Finally, we do not directly transfer VideoMAE parameters but specifically optimize the model for tactile videos through a three-stage training paradigm (Figure 2(b)), with the first stage specifically designing optical flow-guided masking strategy suitable for tactile video characteristics.
> ***
> **Q3: ...the authors should include experimental results that demonstrate the model’s performance in zero-shot scenarios.**
>
> **A3**: In Section A.4.4 of the supplementary materials, we tested cross-sensor robustness by evaluating generalization capability. We used original Tac3D videos with signal patterns significantly different from our training data, and results show **our method maintains strong performance across these heterogeneous modalities.**
>
> We also conducted zero-shot experiments on the XENSE sensor (a variant of the GelSlim sensor [2]). Using the same data collection protocol, we gathered data from 20 objects and generated 100 templates per task with the XENSE sensor. The trained VTV-LLM-7B model was then applied for zero-shot inference. The results in the following table demonstrate that **despite training on only three sensor types, our method generalizes well to other sensors.**
> |                                     | Hardness | Protrusion | Elasticity | Friction | Combined | SFD  | SOI  | OSC  | TSA  | Average |
> |-------------------------------------|----------|------------|------------|----------|----------|------|------|------|------|---------|
> | VTV-LLM-7B (for original 3 sensors) | 73.9     | **75.0**       | **67.3**       | **56.5**     | **35.6**     | 71.3 | **57.6** | **43.2** | 64.0 | **60.4**    |
> | VTV-LLM-7B (for XENSE sensor)       | **77.0**     | 71.0       | 55.0       | 52.0     | 26.0     | **78.0** | 55.0 | 39.0 | **69.0** | 58.0    |
> ***
> **Q4: …the dataset’s limited scale...may restrict its applicability and generalization potential in real-world embodied interaction settings.**
>
> **A4**: Our dataset contains 100 objects with 150,000 visuo-tactile video frames collected systematically using three sensors per object, covering pressing, rotation, and sliding interactions. **This represents the first large-scale dataset specifically designed for visuo-tactile video understanding with comprehensive tactile attribute annotations.**
>
> **Results demonstrate the dataset's effectiveness.** VTV-LLM significantly outperforms all baselines including GPT-4o and excels on untrained tactile analysis tasks. Zero-shot performance drops minimally (60.4% to 58.0% on XENSE sensor) and robotic experiments (Figure 8) confirm strong generalization and practical applicability.
>
> ***
> **Q5: …What is the size after interpolation?...If so, might this approach fail to fully leverage the fine-grained capabilities of Tac3D data in perceiving force?**
>
> **A5**: As described in lines 123-125, we employ cubic spline interpolation to address the resolution disparity between Tac3D (20×20) and other high-resolution tactile sensors, interpolating Tac3D force field data to **320×320 pixels** for training consistency.
>
> Figure 1 shows the interpolated Tac3D data used for model training, while Figure 7 demonstrates zero-shot testing on original 20×20 force field data. This design proves that despite training only on interpolated image-format data, our model successfully generalizes to original force field format and maintains fine-grained force perception capabilities, validating that our approach preserves Tac3D's essential force-sensing characteristics.
> **Converting force data to image format enables unified processing across heterogeneous sensors and provides denser spatial representations that better capture force distributions for our video-based learning paradigm, rather than diminishing force information.**
>
> ***
> **Q6: ..."maintains strong performance across these heterogeneous input modalities" is not sufficiently...**
>
> **A6**: We have added more zero-shot tests in **A3**.
>
> ***
> **Q7: Is it suitable for real-world dexterous manipulation tasks that demand real-time performance, or is it limited to tactile reasoning applications?**
>
> **A7**: While VTV-LLM focuses on tactile reasoning, our robotic experiments (Section A.4.5) demonstrate its real-world potential by successfully performing object selection and manipulation tasks. Although current computational requirements limit real-time performance, our framework provides essential building blocks for future dexterous manipulation systems.
> ***
> [1] Tong, Zhan, et al. "Videomae: Masked autoencoders are data-efficient learners for self-supervised video pre-training." Advances in neural information processing systems 35 (2022): 10078-10093.
>
> [2] Taylor, Ian H., et al. "Gelslim 3.0: High-resolution measurement of shape, force and slip in a compact tactile-sensing finger." 2022 International Conference on Robotics and Automation (ICRA). IEEE, 2022.

---

> > ### Comment · Reviewer_Q6gv · 2025-08-04
> >
> > Thank you for the detailed response. Most of my concerns have been addressed. I believe this work presents incremental improvements on both the model and the dataset, so I will raise my final score to borderline accept.

---

> > > ### Author Response · Authors · 2025-08-05
> > >
> > > Thank you very much for your feedback. We are glad that our rebuttal effectively addressed your concerns. We will incorporate the revised content into the final version.

---

> ### Comment · Area_Chair_HA1p · 2025-08-03
>
> Dear Reviewer,
>
> Could you please check if the authors’ rebuttal adequately addresses your concerns? If so, kindly acknowledge the rebuttal and provide any additional comments. If not, it would be greatly appreciated it if you could engage in a discussion with the authors. Your input at this stage is essential to the review process. Thank you very much for your time and effort!
>
> AC

---

> ### Comment · Area_Chair_HA1p · 2025-08-06
>
> Dear Reviewer,
>
> Please remember to submit the *Mandatory Acknowledgement*. Thank you!
>
> AC

---

### Official Review · Reviewer_5L3e · 2025-07-02

**Clarity:** 4
**Significance:** 3
**Originality:** 4
**Rating:** 5
**Confidence:** 4

**Summary:**

The paper proposes a universal Visuo-Tactile Video Large Language Model(VTV-LLM) to address the gap in integrating tactile perception into LLMs. The authors collect a large-scale tactile video dataset, which contains 150K video frames with 4 fundamental tactile attributes. A three-stage training paradigm is proposed to train the VTV-LLM, which includes a VTV enhancement for robust visuo-tactile representation, a VTV-text alignment for cross-modal correspondence, and a text prompt finetuning for natural language generation. After training, the VTV-LLM can execute various tactile reasoning tasks.

**Questions:**

The annotation focuses on four fundamental tactile attributes. Are there plans to expand the dataset to more complex tactile features, such as texture and temperature?
Is the VTV-LLM possible for edge-device deployment?
How to mitigate the adverse effects of the LLMs' hallucination issue?

**Ethical Concerns:**

["NO or VERY MINOR ethics concerns only"]

**Final Justification:**

Accept

**Limitations:**

The dominating limitation is the real-world feasibility of the LLM-based tactile reasoning model in real-time scenarios.

**Quality:**

3

**Strengths And Weaknesses:**

【Strengths】
1. The first work to integrate visual, tactile, and language modalities for universal tactile understanding.
2. A valuable and comprehensive dataset encompasses a diverse range of objects, sensors, and tactile attributes, which can be used for training tactile-related models.
3. The three-stage paradigm guarantees effective cross-sensor and cross-modal integration, enhancing tactile reasoning accuracy.
4. The trained model performs well on the tactile reasoning task and lays the groundwork for the embodied domain.

【Weaknesses】
Limitation of the dataset: a small number of tactile sensors (GelSight, DIGIT, Tac3D) and limited fundamental attributes.
Real-world feasibility concerns: The computational overhead of the LLM restricted its application in real-time embodied interaction scenarios.

---

> ### Author Rebuttal · Authors · 2025-07-26
>
> We thank reviewer for the constructive comments. We provide our feedbacks as follows.
>
> **Q1: Limitation of the dataset: a small number of tactile sensors (GelSight, DIGIT, Tac3D) and limited fundamental attributes.**
>
> **A1**: Thanks for this comment. For the sensors, **our three selected sensors are the most common commercial sensors, representing the main technological approaches in current visuo-tactile sensing (GelSight and DIGIT for capturing high-resolution information, and Tac3D for measuring deformation force fields), while existing datasets [1][2] typically focus only on single sensor types.** As stated in lines 72-74, these sensors have garnered widespread attention in robotic applications, such as material classification, shape reconstruction, and dexterous manipulation tasks, demonstrating their importance in the visuo-tactile field.
>
> Moreover, our method successfully addresses the challenge of cross-sensor data discrepancies, and the robustness test in Supplementary Material A.4.4 demonstrates good generalization performance across different data distributions, which is itself an important technical contribution.
> To further verify the effectiveness, **we also conducted a zero-shot experiment on the XENSE sensor (a variant of the GelSlim sensor [3]).** In the experimental setup, we use the XENSE sensor to perform the same data collection steps for 20 objects as in the original experiment and generated 100 templates for each task. Then, the trained VTV-LLM-7B model is used for zero-shot inference. Related results can be found in the following table, it proves that **although our method is trained on only three kinds of sensor data, it generalizes well for the other sensors.**
> |                                     | Hardness | Protrusion | Elasticity | Friction | Combined | SFD  | SOI  | OSC  | TSA  | Average |
> |-------------------------------------|----------|------------|------------|----------|----------|------|------|------|------|---------|
> | VTV-LLM-7B (for original 3 sensors) | 73.9     | **75.0**       | **67.3**       | **56.5**     | **35.6**     | 71.3 | **57.6** | **43.2** | 64.0 | **60.4**    |
> | VTV-LLM-7B (for XENSE sensor)       | **77.0**     | 71.0       | 55.0       | 52.0     | 26.0     | **78.0** | 55.0 | 39.0 | **69.0** | 58.0    |
>
> For the limited fundamental attributes, our four selected attributes (hardness, protrusion, elasticity, friction) are based on fundamental theories of tactile perception [4]: **hardness and elasticity for reflecting the mechanical deformation properties of materials, protrusion for capturing surface geometric texture information, and friction for characterizing surface interaction dynamics.** As stated in lines 22-23: "*This tactile feedback enables sophisticated physical reasoning and interaction in our environment,*" our attribute selection covers the core dimensions of tactile perception.
> Moreover, our methodology and technical framework have good scalability and can easily integrate new sensor types and attribute dimensions.
> ***
> **Q2: Real-world feasibility concerns: The computational overhead of the LLM restricted its application in real-time embodied interaction scenarios.**
>
> **A2**: As shown in Figure 5, we provide three different parameter-scale model variants (3B, 7B, 14B), allowing selection based on computational resource constraints of specific application scenarios. **The VTV-LLM-3B model significantly reduces computational overhead while maintaining reasonable performance, providing a feasible solution for resource-constrained robotic platforms.**
>
> Moreover, Supplementary Material A.4.5 demonstrates our deployment testing in real robotic systems, using a UR5 collaborative robot with Robotiq-85 gripper integrated with GelSight Mini sensor. The experiments prove the system's operability in real environments. This practical validation demonstrates that **despite computational overhead, the current system can effectively operate on real robotic platforms.**
>
> Finally, many important embodied interaction applications do not require millisecond-level real-time response, including material property assessment and quality control. As the first work in this field, our main contribution lies in demonstrating the feasibility of visuo-tactile-language multimodal understanding and establishing an evaluation benchmark for visuo-tactile video.
>
> ***
> **Q3: The annotation focuses on four fundamental tactile attributes. Are there plans to expand the dataset to more complex tactile features, such as texture and temperature? Is the VTV-LLM possible for edge-device deployment? How to mitigate the adverse effects of the LLMs' hallucination issue?**
>
> **A3**: Our four selected fundamental attributes (hardness, protrusion, elasticity, friction) cover the core dimensions of tactile perception. As described in lines 133-139, these attributes are systematically categorized into three levels, providing a comprehensive foundation for tactile attribute analysis in downstream reasoning tasks. We plan to expand to more physical properties in the next stage of work, such as temperature, sharpness and so on.
>
> Figure 5 shows our three model scales (3B, 7B, 14B). The VTV-LLM-3B model significantly reduces parameter count while maintaining reasonable performance, making edge deployment possible.
>
> For the hallucination issue, our three-stage training paradigm (Figure 2(b)) establishes reliable cross-modal correspondence by gradually aligning visuo-tactile representations with textual descriptions, reducing the possibility of hallucination generation. Moreover, unlike open-ended image description, our tasks are based on clearly defined physical properties (hardness, protrusion, elasticity, friction). **This structured output space naturally limits the scope of hallucination generation.**
> ***
> [1] Cheng, Ning, et al. "Touch100k: A large-scale touch-language-vision dataset for touch-centric multimodal representation." Information Fusion (2025): 103305.
>
> [2] Yu, Samson, et al. "Octopi: Object Property Reasoning with Large Tactile-Language Models." Robotics: Science and Systems. 2024.
>
> [3] Taylor, Ian H., et al. "Gelslim 3.0: High-resolution measurement of shape, force and slip in a compact tactile-sensing finger." 2022 International Conference on Robotics and Automation (ICRA). IEEE, 2022.
>
> [4] Stefani, Antonio Luigi, et al. "Signal processing for haptic surface modeling: A review." Signal Processing: Image Communication (2025): 117338.

---

> ### Comment · Area_Chair_HA1p · 2025-08-03
>
> Dear Reviewer,
>
> Could you please check if the authors’ rebuttal adequately addresses your concerns? If so, kindly acknowledge the rebuttal and provide any additional comments. If not, it would be greatly appreciated it if you could engage in a discussion with the authors. Your input at this stage is essential to the review process. Thank you very much for your time and effort!
>
> AC

---

> ### Author Response · Authors · 2025-08-05
> **To Reviewer 5L3e**
>
> Dear Reviewer 5L3e,
>
> Thanks again for your diligent effort in reviewing our submission. We have carefully addressed the concerns raised. As the discussion phase deadline is approaching, we sincerely hope you can consider positively recommending our work if your concerns are solved. If you still have further comments/suggestions, please don't hesitate to let us know.
>
> Best regards, Authors

---

### Official Review · Reviewer_P8wL · 2025-07-07

**Clarity:** 3
**Significance:** 1
**Originality:** 2
**Rating:** 2
**Confidence:** 4

**Summary:**

This paper introduces the Visuo-Tactile Video (VTV) dataset, which comprises 150K frames of tactile videos annotated with attributes. Additionally, the author proposes the VTV-LLM framework, which facilitates reasoning in embodied interactions. To train the VTV-LLM framework, the author utilizes the VTV dataset and introduces a three-stage training process for the model.

**Questions:**

Please refer to weakness section.

**Ethical Concerns:**

["NO or VERY MINOR ethics concerns only"]

**Limitations:**

Yes

**Quality:**

2

**Strengths And Weaknesses:**

**Strength**
1. Introductin of VTV dataset that include attribute annotation and tactile Q&A
2. Demonstrates the effectiveness of the proposed VTV-LLM and three-step training strategy through experiments on tactile understanding tasks.

**Weakness**
1. I fail to comprehend the advantage of video over tactile. What is the advantage of tactile video compared to tactile images? I believe tactile video doesn’t offer significant differences over time, potentially resulting in redundant frames.

2. Additionally, since the network processes video as input, the number of videos becomes more crucial than the number of frames in the dataset. Furthermore, it’s unclear whether annotations are applied to videos or frames. For instance, what does 150,000 video frames represent? I think including frames per second (FPS) would help understand the dataset size or the number of videos. 150,000 frames at 60 FPS could equate to just 40 minutes of video, which is not a substantial amount.

3. For VTV-LLM, it’s unclear why tactile video is necessary for tactile understanding. A single image would suffice for the task proposed by the task author. Video input only increases the model’s size. To demonstrate the necessity of video, a task that shows temporal tactile information should be introduced.

---

> ### Author Rebuttal · Authors · 2025-07-27
>
> We thank reviewer for the constructive comments. We provide our feedbacks as follows.
>
> **Q1: I fail to comprehend the advantage of video over tactile. What is the advantage of tactile video compared to tactile images? I believe tactile video doesn’t offer significant differences over time, potentially resulting in redundant frames.**
>
> **A1**: Thanks for this comment. Regarding your question about the advantages of tactile video over tactile images, I believe there is a conceptual misunderstanding that needs clarification. Tactile videos indeed provide crucial temporal information in physical interactions that static tactile images cannot capture, which is essential for understanding dynamic tactile attributes of objects.
>
> First, our research design includes three sequential interaction modes: normal pressing, rotational movement, and sliding motion. This design is not arbitrary but based on the physical principles of tactile perception. As clearly described in lines 129-131 of our paper: *"Our data collection process consisted of three sequential interactions: (1) normal pressing against the object surface to capture pressure distribution, (2) rotational movement to acquire shear information, and (3) sliding motion to obtain friction characteristics."* Each interaction mode reveals different physical properties of objects: pressing reveals hardness and elasticity, rotation captures shear characteristics, and sliding obtains friction information. **These dynamic interaction processes cannot be represented in a single static image.**
>
> More importantly, **tactile videos can capture the deformation patterns and recovery behaviors of objects during interactions.** For example, when we press an elastic object, its surface deforms and gradually returns to its original state after release. This temporal change pattern directly reflects the object's elastic properties. As stated in lines 41-43 of our paper: *"the temporal dimension of tactile interactions, which captures how surfaces respond to pressing, sliding, and rotational movements, remains underexplored in current approaches, despite containing crucial information about dynamic material attributes."* **This temporal dimension contains crucial dynamic material attribute information that static methods cannot acquire.**
>
> From a technical implementation perspective, our proposed optical flow-guided masking method is specifically optimized for the motion characteristics of tactile videos. Lines 186-191 of our paper explain why directly applying existing tube masking methods would be problematic: *"While VideoMAE offers a powerful masked video autoencoder... this approach assumes minimal motion across large frame regions. This assumption proves problematic for visuo-tactile videos, which typically exhibit significant motion patterns. Direct application of tube masking to such inputs risks substantial information leakage."* **Our method, through dense motion estimation and backward warping techniques, can better handle significant motion patterns in tactile videos, which is a technical challenge that doesn't need to be considered when processing static images [1].**
>
> Experimental results also confirm the importance of temporal information. Our ablation study (Table 2) shows that using our temporally-aware method achieves a 36.1% improvement in average performance over the VideoMAE baseline without training (from 22.9% to 59.0%). This significant performance improvement directly demonstrates the value of temporal information in tactile videos. **If tactile videos were truly just collections of redundant frames, we would not observe such significant performance improvements.**
>
> Finally, from an application perspective, real robotic interactions require understanding how objects behave during dynamic contact processes. Static tactile images can only provide contact information at a specific moment and cannot guide robots on how to adjust grasping force or manipulation strategies. **The dynamic feedback provided by tactile videos enables robots to adjust their operational behavior in real-time, which is indispensable for complex physical interaction tasks [2].** Our VTV-LLM is designed precisely to address this need for dynamic tactile understanding and natural language interaction.
> ***
> **Q2: Additionally, since the network processes video as input, the number of videos becomes more crucial than the number of frames in the dataset. Furthermore, it’s unclear whether annotations are applied to videos or frames. For instance, what does 150,000 video frames represent? I think including frames per second (FPS) would help understand the dataset size or the number of videos. 150,000 frames at 60 FPS could equate to just 40 minutes of video, which is not a substantial amount.**
>
> **A2**: Thank you for this important clarification question regarding our dataset specifications. We appreciate the opportunity to provide more detailed information about VTV150K.
>
> **Our annotations are applied directly to videos, not individual frames.** Each video represents a complete tactile interaction sequence of 5 seconds, capturing the full temporal dynamics of object contact. The 150,000 video frames represent the total frames across all videos in our dataset, with each video recorded at **20 FPS** and a resolution of **320×320 pixels**. This means our dataset contains **1,500 complete tactile interaction videos** (150,000 frames / 100 frames per video = 1,500 videos), providing **125 minutes** of total video content focused on tactile understanding.
>
> **When comparing VTV150K to existing tactile-text annotated datasets [3] [4], our dataset represents the largest collection of visuo-tactile video frames with natural language annotations currently available.** The Touch100k dataset [3] contains 100,147 touch-language-vision entries but focuses primarily on static tactile properties with limited temporal dynamics. The PHYSICLEAR dataset [4] includes 408 tactile videos from 74 objects using only GelSight sensors, which is significantly smaller in scale than our 1,500 videos from 100 objects.
>
> Moreover, a crucial distinction of VTV150K is its multi-sensor coverage. **While existing datasets [3][4] typically focus on single sensor types, our dataset incorporates three widely-used commercial tactile sensors: GelSight Mini, DIGIT, and Tac3D.** This multi-sensor approach addresses a critical gap in the field, as real-world tactile applications often require cross-sensor generalization. The diversity across sensor types, combined with our comprehensive object selection (100 diverse everyday objects), provides a more robust foundation for developing universal tactile understanding models.
>
> Finally, the video-based nature of our dataset is particularly important for capturing dynamic tactile attributes like elasticity and friction, which cannot be adequately represented through static frames alone. **Each 5-second video at 20 FPS provides sufficient temporal resolution to capture the complete interaction cycle including initial contact, deformation, and recovery patterns essential for understanding material properties.** This temporal richness distinguishes VTV150K from frame-based approaches and enables learning of sophisticated tactile dynamics that are crucial for embodied interaction tasks.
> ***
> **Q3: For VTV-LLM, it’s unclear why tactile video is necessary for tactile understanding. A single image would suffice for the task proposed by the task author. Video input only increases the model’s size. To demonstrate the necessity of video, a task that shows temporal tactile information should be introduced.**
>
> **A3**: Thank you for this comment. In fact, the question raised in **Q3** is similar to that in **Q1**, both focusing on the necessity of tactile video understanding. For specific explanations, please refer to **A1**.
>
> Moreover, to further demonstrate the value of temporal modeling in tactile perception, we adopted Octopi [4]'s data reading method (selecting 5 frames) and trained using the identical dataset and pipeline. **The results shown in the table below demonstrate the importance of tactile video understanding, especially for dynamic attributes.**
>
> |                    | Hardness | Protrusion | Elasticity | Friction | Combined | SFD  | SOI  | OSC  | TSA  | Average |
> |--------------------|----------|------------|------------|----------|----------|------|------|------|------|---------|
> | Selecting 5 frames | 62.6     | 60.7       | 41.3       | 31.7     | 10.8     | 58.1 | 47.5 | 27.1 | 51.0 | 43.4    |
> | Ours               | **73.9**     | **75.0**       | **67.3**       | **56.5**     | **35.6**     | **71.3** | **57.6** | **43.2** | **64.0** | **60.4**    |
>
> ***
> [1] Zhao, Jialiang, et al. "Transferable Tactile Transformers for Representation Learning Across Diverse Sensors and Tasks." Conference on Robot Learning. PMLR, 2025.
>
> [2] Zhao, Zihang, et al. "Embedding high-resolution touch across robotic hands enables adaptive human-like grasping." Nature Machine Intelligence (2025): 1-12.
>
> [3] Cheng, Ning, et al. "Touch100k: A large-scale touch-language-vision dataset for touch-centric multimodal representation." Information Fusion (2025): 103305.
>
> [4] Yu, Samson, et al. "Octopi: Object Property Reasoning with Large Tactile-Language Models." Robotics: Science and Systems. 2024.

---

> ### Comment · Area_Chair_HA1p · 2025-08-03
>
> Dear Reviewer,
>
> Could you please check if the authors’ rebuttal adequately addresses your concerns? If so, kindly acknowledge the rebuttal and provide any additional comments. If not, it would be greatly appreciated it if you could engage in a discussion with the authors. Your input at this stage is essential to the review process. Thank you very much for your time and effort!
>
> AC

---

> ### Author Response · Authors · 2025-08-05
> **To Reviewer P8wL**
>
> Dear Reviewer P8wL,
>
> Thanks again for your diligent effort in reviewing our submission. We have carefully addressed the concerns raised and conducted the requested experiments. As the discussion phase deadline is approaching, we sincerely hope you can consider positively recommending our work if your concerns are solved. If you still have further comments/suggestions, please don't hesitate to let us know.
>
> Best regards, Authors

---

### Decision · Program_Chairs · 2025-09-17

**Decision:**

Accept (poster)

**Comment:**

This work introduces the first multimodal large language model that supports universal visuo-tactile video understanding. The authors collect a large-scale tactile video dataset with annotated attributes and propose a three-stage training paradigm that enables the model to achieve superior performance on tactile reasoning tasks.

Most reviewers consider the proposed dataset valuable for future research in the field. They also appreciate the approach’s design and the demonstrated empirical improvements. One concern was raised regarding the limited fundamental attributes and granularities, but this was well explained during the rebuttal. Other minor concerns from reviewers, except for Reviewer P8wL, who did not participate in the rebuttal or discussion, were also addressed by the authors.

Given the above, the AC recommends acceptance. This decision is made based on down-weighting the comments from Reviewer P8wL due to their lack of participation and discussion.